# Prediction of fluid intelligence from T1-w MRI images: A precise two-step deep learning framework

**Mingliang Li**[1,2], **Mingfeng Jiang**[1]*, **Guangming Zhang**[1], **Yujun Liu**[1], **Xiaobo Zhou**[3]*

**1** West China Biomedical Big Data Center, West China Hospital, West China School of Medicine, Sichuan University, Chengdu, China, **2** Med-X Center for Informatics, Sichuan University, Chengdu, China, **3** Center for Computational Systems Medicine, School of Biomedical Informatics, University of Texas Health Science Center at Houston, Houston, TX, United States of America

* 17010230513@gznu.edu.cn (MJ); zhouxb2015@163.com (XZ)

**Data Availability Statement:** Data were obtained from the NIMH Data Archive (NDA) database, generated by the Ado-lescent Brain Cognitive Development (ABCD) study, the largest long-term study of brain development and child health in the

## Abstract

The Adolescent Brain Cognitive Development (ABCD) Neurocognitive Prediction Challenge (ABCD-NP-Challenge) is a community-driven competition that challenges competitors to develop algorithms to predict fluid intelligence scores from T1-w MRI images. In this work, a two-step deep learning pipeline is proposed to improve the prediction accuracy of fluid intelligence scores. In terms of the first step, the main contributions of this study include the following: (1) the concepts of the residual network (ResNet) and the squeeze-and-excitation network (SENet) are utilized to improve the original 3D U-Net; (2) in the segmentation process, the pixels in symmetrical brain regions are assigned the same label; (3) to remove redundant background information from the segmented regions of interest (ROIs), a minimum bounding cube (MBC) is used to enclose the ROIs. This new segmentation structure can greatly improve the segmentation performance of the ROIs in the brain as compared with the classical convolutional neural network (CNN), which yields a Dice coefficient of 0.8920. In the second stage, MBCs are used to train neural network regression models for enhanced nonlinearity. The fluid intelligence score prediction results of the proposed method are found to be superior to those of current state-of-the-art approaches, and the proposed method achieves a mean square error (MSE) of 82.56 on a test data set, which reflects a very competitive performance.

## 1 Introduction

Understanding cognitive development in children may potentially improve their health outcomes through adolescence. Thus, determining the neural mechanism underlying general intelligence is a critical task. Fluid intelligence is one crucial component of general human intelligence, which involves the capacity to think logically and solve problems in novel situations and is independent of acquired knowledge [1]. It has been widely accepted that fluid intelligence reaches a peak in late adolescence, after which it declines. Thus, its quantification and accurate prediction are important for teenagers, as it foresees creative achievement,

United States. Information about the ABCD Data Repository can be found at https://nda.nih.gov/abcd/about.

**Funding:** This research was supported by Center of Excellence-International Collaboration Initiative Grant (Grant Number: 139170052), 1.3.5 project for disciplines of excellence, West China Hospital, Sichuan University (Grant Number: ZYJC18010). The funders had no role in study design, data collection and analysis, decision to publish, or preparation of the manuscript

**Competing interests:** The authors have declared that no competing interests exist.

scholastic performance, employment prospects, socioeconomic status, etc., in their future years. Structural and functional magnetic resonance imaging (MRI) images are one of the most powerful tools to help predict fluid intelligence. Aiming at the precise prediction of fluid intelligence scores, the ABCD dataset provides data and MRI images of a large number of adolescent participants, which have been adjusted for different data collection sites, demographic variables, and whole brain volumes.

The study of fluid intelligence has traditionally been concerned with the identification of the underlying mechanism responsible for cognitive ability. The research results indicate a strong correlation between brain volume and intelligence, and the magnitude of this effect is likely large [2]. More recently, MRIs have been shown to contain useful structural information with a strong correlation to fluid intelligence [3]. In the most related work, the reference [4] has outlined machine learning approaches employed to predict fluid intelligence from brain MRI data.

The traditional method for the prediction of fluid intelligence scores is to calculate features extracted with the assistance of existing computer-aided tools, and then to train a machine learning model on these features. FreeSurfer extracts the volume and thickness features describing the brain structure, which can provide more information for the prediction of fluid intelligence scores [5]. The National Consortium on Alcohol and Neurodevelopment in Adolescence (NCANDA) pipeline can be used to complete brain image noise reduction, correction, and feature extraction [6]. Moreover, the subcortical regions of subjects have been segmented by FSL FIRST; these regions were mainly cortical and did not include any subcortical regions of interest (ROIs) [7]. Furthermore, brain global shape features have been calculated via the implementation of the Insight Segmentation and Registration Toolkit (ITK) [8].

In recent years, deep learning methods have emerged as state-of-the-art solutions to many problems spanning various domains, such as natural language processing, bioinformatics, and medical imaging [9]. The convolutional neural network (CNN), a type of deep learning model, has been a useful tool for the analysis of image data [10]. Some studies have utilized structural MRI images to predict fluid intelligence scores, and brain volume has been demonstrated to be related to quantitative reasoning and working memory [11]. Moreover, a novel framework has been proposed for the estimation of a subject's intelligence score via sparse learning based on neuroimaging features [12].

The use of traditional deep learning methods for fluid intelligence score regression is characterized by the following weaknesses: (1) the structure of the segmentation model is not well optimized: first, the high-layer and low-layer features of the segmentation framework are not fused [13]. As a result, a substantial amount of spatial information in the image is lost in the lower layer; second, the attention mechanism is not introduced into the segmentation model. The attention mechanism guides segmentation model by giving the higher weight to focus features while minimizing the irrelevant features, giving them lower weights; (2) the segmentation results are not used to create a neural network; instead, machine learning methods are used [14]. Thus, the traditional methods cannot fit the intelligence score well, resulting in unsatisfactory prediction accuracy.

In the present work, T1-weighted MRI images of adolescents are utilized to predict their fluid intelligence scores with a novel precise two-step deep learning framework. The main contributions of this work are three-fold: (1) residualized fluid intelligence scores are predicted based on an improved 3D U-Net architecture that utilizes the concepts of the residual network (ResNet), squeeze-and-excitation network (SENet), and symmetry learning mechanism; (2) the pixels in symmetrical brain regions share the same label, and the minimum bounding cube (MBC) operation is employed to eliminate interference from the background; (3) more

accurate and stable results are obtained via fine segmentation, and these results are more help-ful for improving the prediction accuracy of fluid intelligence scores.

## 2 Data

### 2.1 Dataset

Data were provided by the 2019 ABCD Neurocognitive Prediction Challenge (ABCD-NP--Challenge) [15], and included data on children aged 9–10. Participants were given access to T1-weighted MRI scans from 3739 children for training, scans from 415 children for valida-tion, and scans from 4515 children for testing. The fluid intelligence scores recorded by the ABCD study were measured via the NIH Toolbox Neurocognition battery, as detailed in the electronic S1 File (ESI). To minimize the impact of confounds that are not related to the brain structure, the raw scores were pre-residualized by the ABCD-NP- Challenge organizers based on sex at birth, ethnicity, highest parental education, parental income, parental marital status, brain volume, and image acquisition site.

3D T1-w MRI images were pre-processed by the challenge organizers. The pre-processing steps involved first transforming raw data into NIfTI formats [16]. The brain mask was created by a majority voting approach among the outputs of a series of neuroimaging software includ-ing FSL BET, AFNI 3dSkullStrip, FreeSurer mri gcut, and Robust Brain Extraction (ROBEX). The final mask was obtained by taking a majority voting of the resulting masks, and it removed noise and corrected for bias-field inhomogeneities. Based on the final masks, the T1-w MRI image was segmented into gray matter, white matter, and cerebrospinal fluid via Atropos [17]. Afterwards, the skull-stripped T1-w image and corresponding gray matter segmentations were affinely mapped to the SRI24 atlas [18].

Finally, 122 brain regions of interest (ROIs) extracted by the challenge organizers based on the SRI24 atlas. 14 brain ROIs with unique anatomical characteristics and the roles in cognitive functions were selected to predict fluid intelligence score, as specifically described in the next section.

### 2.2 Selected brain regions

Most of the 14 ROIs for analysis by the proposed method have previously been reported and are highly associated with cognitive ability, as shown in Table 1. It has been found that the hip-pocampus, an important component in the limbic system, plays an important role in memory and spatial navigation, and the thalamus is conceptualized as a switchboard of information that processes and relays sensory information[1, 19, 20]. The inferior frontal gyrus has also been found to be related to semantic task processing. Moreover, recent novel views of thalamic functions emphasize integrative roles in cognition, ranging from learning and memory to flex-ible adaption [21]. The caudate nucleus is related to cognitive tasks such as organizing behav-ioral responses and using verbal skills in problem-solving [22]. Considerable evidence suggests that the human amygdala plays an important role in higher cognitive functions in addition to its well-known role in emotional processing [23].

In anatomy, the frontal gyrus has six regions, including L inferior frontal gyrus—opercular, R inferior frontal gyrus—opercular, L inferior frontal gyrus—triangular, R inferior frontal gyrus—triangular, L inferior frontal gyrus—orbital, and R inferior frontal gyrus—orbital; the hippocampus has two regions, including L hippocampus and R hippocampus; the amygdala has two regions, including L amygdala and R amygdala; the caudate nucleus has two regions, including L caudate nucleus and R caudate nucleus; the thalamus has two regions, including L thalamus and R thalamus.

**Table 1. The labels and names of the ROIs in the SRI24 space.**

| ROIs | Label | Name |
|---|---|---|
| L Inferior frontal gyrus—opercular | 11 | Frontal gyrus |
| R Inferior frontal gyrus—opercular | 12 | |
| L Inferior frontal gyrus—triangular | 13 | |
| R Inferior frontal gyrus—triangular | 14 | |
| L Inferior frontal gyrus—orbital | 15 | |
| R Inferior frontal gyrus—orbital | 16 | |
| L Hippocampus | 37 | Hippocampus |
| R Hippocampus | 38 | |
| L Amygdala | 41 | Amygdala |
| R Amygdala | 42 | |
| L Caudate nucleus | 71 | Caudate nucleus |
| R Caudate nucleus | 72 | |
| L Thalamus | 77 | Thalamus |
| R Thalamus | 78 | |

L/R indicates a location in the left/right hemisphere.

## 3 Methodology

### 3.1 Symmetry learning mechanism

The vertebrate cerebrum (brain) is formed by two cerebral hemispheres that are separated by a groove, namely the longitudinal fissure. The brain can thus be described as being divided into left and right cerebral hemispheres. Macroscopically, the hemispheres are roughly mirror images of each other, with only subtle differences. On the microscopic level, the cytoarchitecture of the cerebral cortex reveals the functions of cells, the quantities of neurotransmitter levels, and receptor subtypes to be markedly asymmetrical between the hemispheres [24, 25]. However, while some of these hemispheric distribution differences are consistent across human beings, many vary from individual to individual [26]. It is precisely because the hemispheres are roughly mirror images of each other that the pixels in symmetrical brain regions were assigned the same label in the segmentation process. The ROIs presents macroscopic symmetry, the specific details are presented in Fig 1.

### 3.2 Technical details

The traditional pipeline of the regression of brain MRI images and fluid intelligence scores is based on the original deep learning architecture and post-processing. First, in the pre-processing stage, the segmentation framework needs to be further optimized. Second, the post-processing stage only uses the median predicted scores as the final prediction result, or extracts the high-level feature map information for regression, which causes information loss in the pre-processing stage. Finally, the fluid intelligence score regression operation is performed for only one ROI at a time, which ignores the interaction between other brain areas [13].

To improve the prediction accuracy of fluid intelligence scores, a two-step deep learning network is proposed. This network was inspired by the original 3D U-Net architecture, ResNet, SENet, and the symmetry learning mechanism. The improved 3D U-Net can perform accurate 3D segmentation tasks, after which the fluid intelligence score is predicted based on the feature of each finely segmented brain ROI.

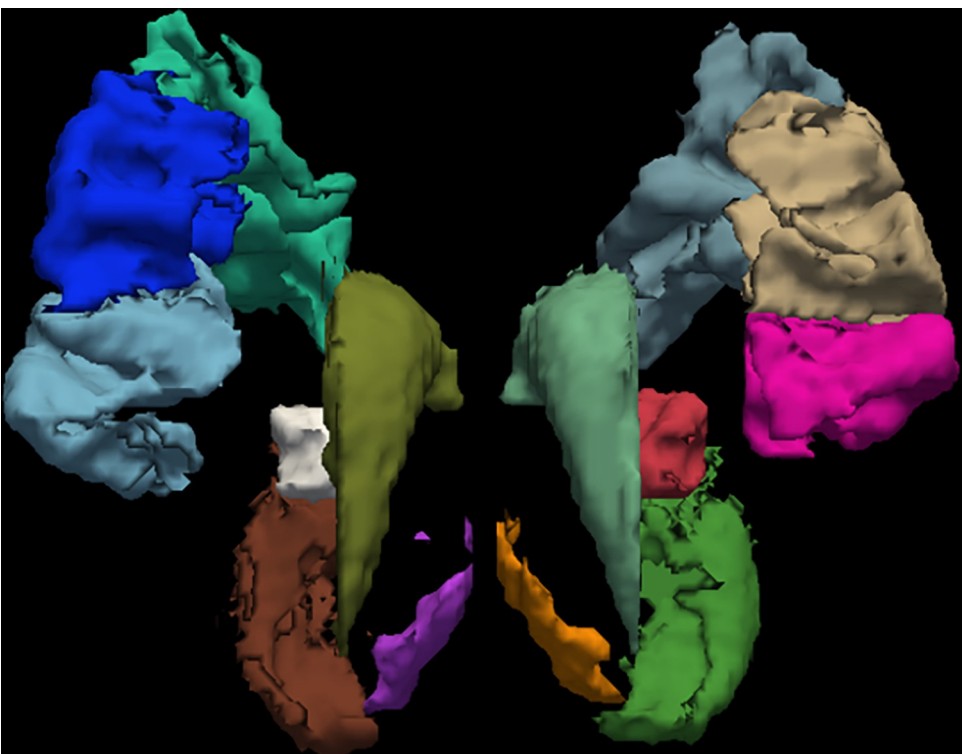

**Fig 1. Example segmentation of the ROIs.**

**A. Segmentation stage.** The brain MRIs of subjects have high similarity, and the original 3D U-Net is able to extract a large number of features. However, due to the existence of individual differences, it is necessary to further enhance the attention mechanism of the network to obtain more refined segmentation results.

While deeper networks can extract more structural information, they lose more local information due to the continuous reduction of the feature map resolution. The architecture of the proposed network is illustrated in Fig 2, and consists of four layers including the bottleneck. It is assumed that the use of four layers is sufficient to extract more location information. The motivation behind the proposed architecture is to improve the attention mechanism.

Therefore, skip connections in the bottleneck between the encoder and decoder layers, the recommendation block, and the SegS-E block are added to the network architecture to improve its ability to capture spatial and spectral *information*; more details are comprehensively provided in the next subsection.

The encoder takes a 3D input patch with size of $112 \times 112 \times 112$ from the set of input images. In the first layer, the 16-channel $112 \times 112 \times 112$ feature maps are generated with a $1 \times 1 \times 1$ convolution operation, and 32-channel $112 \times 112 \times 112$ feature maps are generated with the subsequent recombination block operation. The number of feature maps is increased in the subsequent layers to learn the deep features, which is followed by max-pooling and the down-sampling of features in the encoding layer. To match the size of feature maps in the channel, the 64-channel $28 \times 28 \times 28$ feature maps after the $1 \times 1$ convolution operation are transformed into 96-channel $28 \times 28 \times 28$ feature maps. The skip performs an element-wise addition operation ($\oplus$) in the selected channel to ensure the volumes at this addition operation are the same size. Similarly, in the decoding layer, the feature maps are upsampled. In the

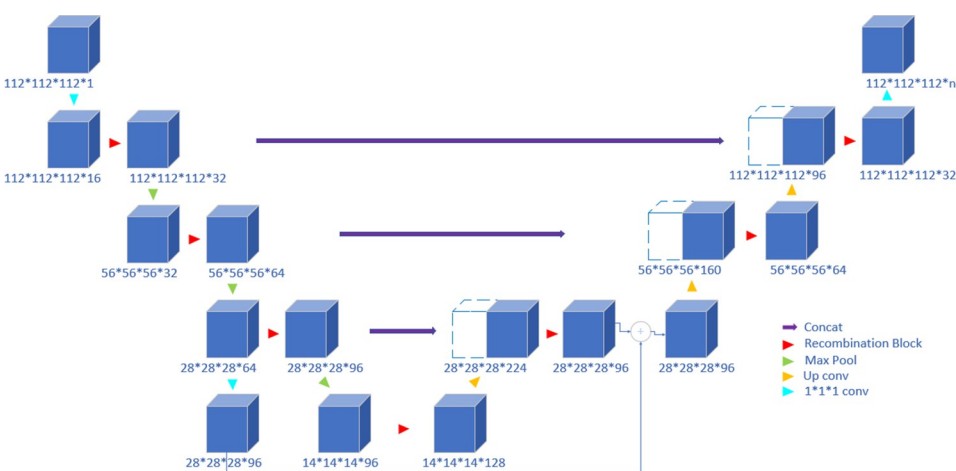

**Fig 2. The improved 3D U-Net architecture.** Blue boxes represent feature maps, and the number of channels is denoted after the size of each feature map. A skip connection is included in the bottleneck between the encoder and decoder layers. ⊕ refers to element-wise addition in the selected channel. The predicted labels are compared with the ground truth to calculate the Dice loss.

output layer, the segmentation map predicted by the model is compared with the corresponding ground truth, and the error is backpropagated.

*1) Recombination block.* The concepts of ResNet and SENet are referenced to construct the recombination block.

With ResNet, the gradients can flow backward directly through the skip connections from the later layers to the initial filters, which can effectively reduce model overfitting [27]. The recombination block aims to enhance the semantic information between different feature layers; primarily, more convolutions and nonlinear transformations are performed so that the model can adapt to its own structure during training.

In the recombination block, the convolution, batch normalization (BN), and SegS-E block feature extraction modules are used. For the convenience of explanation, the labels $B1$-$B_8$ are denoted on the blue cube representing the feature map shown in Fig 3. The SegS-E block in the recombination block is specifically introduced in the next section.

In the present work, the recombination block is formally defined as follows.

$$B_8 = F\{B_1, B_2, \ldots, B_6\} + B_7. \tag{1}$$

In the following notation, $B_1 \in R^{D \times H \times W \times C}$ and $B_8 \in R^{D \times H \times W \times C'}$ are respectively used to denote the input and the output of the reco $R^{D \times H \times W \times C'}$ mbination block. Moreover, ⊕ refers to the addition of two matrices of the same dimension, in which each element is the sum of the original two matrix elements.

The function $F\{\ldots\}$ represents the residual mapping to be learned, specifically, the convolution operations on the feature maps. Convolution operations are performed on $B_1$ with $1 \times 1 \times 1$ convolution kernels, resulting in the feature map $B_2 \in R^{D \times H \times W \times 4C}$ with $4C$ channels. $B_3 \in R^{D \times H \times W \times 4C}$ is obtained by performing a $3 \times 3 \times 3$ convolution and batch normalization operation. Then, $B_5 \in R^{D \times H \times W \times 4C}$ is obtained by performing the SegS-E block operation on $B_4$, and the specific definition of the SegS-E block is provided in the next subsection. Finally, $B_6 \in R^{D \times H \times W \times C'}$ is obtained by performing a $1 \times 1 \times 1$ convolution operation, and $B_7 \in R^{D \times H \times W \times C'}$ is similarly obtained by performing a $1 \times 1 \times 1$ convolution operation.

*2) SegS-E block.* Via the introduction of the attention mechanism, useful features can be captured more accurately. In the proposed method, a more fine-grained feature enhancement

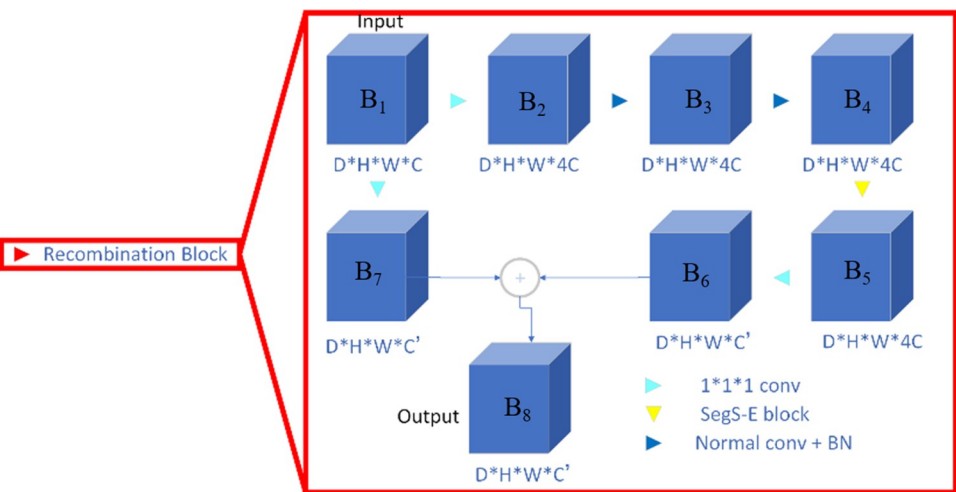

**Fig 3. The recombination block.** "Normal conv" refers to $3 \times 3 \times 3$ convolution filters and batch normalization; "$1 \times 1 \times 1$ conv" refers to $1 \times 1 \times 1$ convolution filters.

method with different weights at different locations is employed, as shown in Fig 4. The attention mechanism of squeeze-and-excitation network (SENet) is used for the SegS-E block [28]. Dilated convolutions "inflate" the kernel by inserting holes between the kernel elements. Dilated convolution is utilized in the SegS-E block, and can systematically aggregate the contextual information of the input. With this purpose, it has applications concerned with the integration of knowledge in the wider context with less cost while keeping the output resolutions high. The goal is to increase the sensitivity of the network by explicitly modeling the channel interdependencies via the use of gated networks. Consequently, the SegS-E block learns how to understand the importance of each feature map in the stack of all the feature maps extracted after a convolution operation, and recalibrates that output to reflect that importance before passing the information to the next layer.

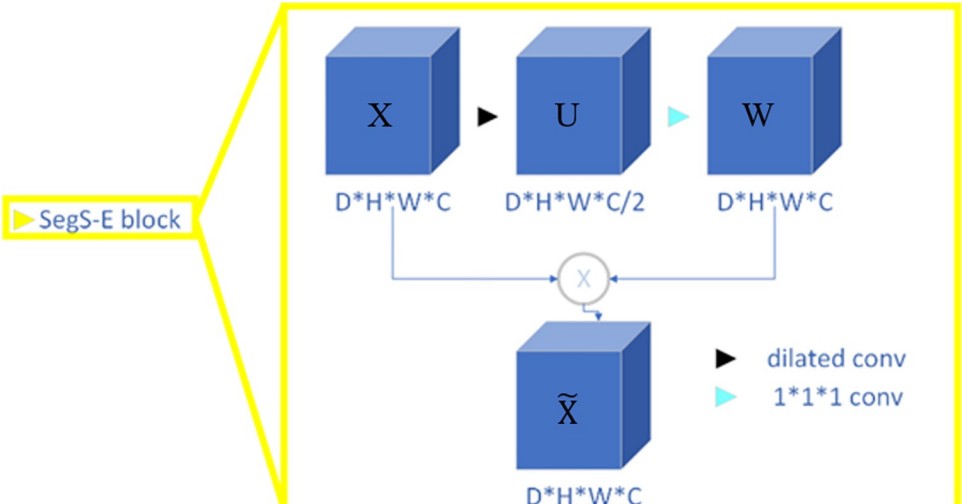

**Fig 4. The SegS-E block.** "Dilated conv" refers to $3 \times 3 \times 3$ kernel with dilation rate of 2, and "$1 \times 1 \times 1$ conv" refers to $1 \times 1 \times 1$ convolution filters.

The details of the SegS-E block are as follows. First, the size of the input feature map size is $D \times H \times W$, and the number of channels is $C$. A dilated convolution with a $3 \times 3 \times 3$ kernel and dilation rate of 2 is then performed over each channel of the input feature maps. Next, a $1 \times 1 \times 1$ kernel is used to perform channel-wise convolution, resulting in a set of feature maps with $C$ channels. Finally, the elements in the matrix of the feature map are multiplied by the elements in the matrix of the processed feature map, and the products are added.

A SegS-E block is a squeeze-and-excitation computational unit, which uses 3D dilation and a $1 \times 1 \times 1$ convolution to map an input $X \in R^{D \times H \times W \times C}$ to feature maps $W \in R^{D \times H \times W \times C}$, after which a Hadamard product operator is used to map $X$ and $W$ to $\tilde{X} \in R^{D \times H \times W \times C}$ as the output. In the following notation, the dilated convolution operation is considered, and $V = [v_1, v_2, \ldots, v_c]$ is used to denote the learned set of filter kernels, where $v_c$ refers to the parameters of the $c$-th dilated convolution filter. Thus, $U$ can be defined as $U = [u_1, u_2, \ldots, u_c]$, where

$$\mathrm{u}_c = \mathrm{v}_c *_l \mathrm{X} = \sum_{s=1}^{c/2} \mathrm{v}_c *_l \mathrm{x}^s, \tag{2}$$

where $*_l$ denotes dilated convolution, and $v_c$ is a 3D kernel that acts on the corresponding channel of $X$. Subsequently, a convolution operation is performed on $U$ with $c$ $1 \times 1 \times 1$ convolution kernels, resulting in feature maps $W \in R^{D \times H \times W \times C}$ with $C$ channels, and $W = [w_1, w_2, \ldots, w_c]$. To systematically aggregate the contextual information of the input, a Hadamard product operation is used to map $X$ and $W$ to $\tilde{X} \in R^{D \times H \times W \times C}$ as the output. The final output of the SegS-E block is

$$\tilde{\mathrm{x}} = \mathrm{W} \otimes \mathrm{X} = \sum_{1}^{c} \mathrm{w}_c \otimes \mathrm{x}_c, \tag{3}$$

where $\tilde{X} = [\tilde{x}_1, \tilde{x}_2, \ldots, \tilde{x}_c]$, and $\otimes$ refers to the Hadamard product. In mathematics, the Hadamard product is the product of two matrices of the same dimensions and has the same dimension as the operands, in which each element is the product of the elements of the original two matrices.

**B. Regression stage.** In the section 3.2-A, we have described the segmentation framework. In this section, we first define the Minimum bounding cube (MBC) of the segmented ROI. Then, we describe the process of building a neural network with *resized* MBCs as input to predict fluid intelligence scores.

1. *Minimum bounding cube (MBC).* To improve the prediction accuracy of fluid intelligence score, we have performed the minimum bounding cube (MBC) operations on segmented ROIs in place of the traditional resizing operation. The details of generating an MBC are as follows: (1) the minimum bounding boxes (MBB) for ROI are generated [29]; (2) $L_{max}$, the longest edge of the MBB of ROI, is determined; (3) the MBB of ROI is resized to fit size of $L_{max} \times L_{max} \times L_{max}$; (3) The MBB with size of $L_{max} \times L_{max} \times L_{max}$ will be resized into a cube of a certain size again, resulting in the minimum bounding cube (MBC). In fact, MBC can be obtained by two interpolation operations, while traditional resizing tasks only need to perform one interpolation operation. In this work, the input size of the regression model is $64 \times 64 \times 64$.

2. *Neural network construction.* In the second stage, to explore the relationship between brain MRI volumes by incorporating morphological information and fluid intelligence scores, a convolutional neural network (CNN) was constructed to map each subject to the corresponding fluid intelligence score based on the ROI segmentation [30].

**Fig 5. Creating a neural network to predict fluid intelligence score.** "Flatten" operation is converting the data into a 1-dimensional array for inputting it to the next layer, and "Conv($3 \times 3 \times 3$) + BN + ReLU" refers to $3 \times 3 \times 3$ convolution filters, BN for batch normalization, and ReLU for rectified linear unit.

The greatest advantage of deep learning algorithms as compared with traditional machine learning models is that they try to learn high-level features from data in an incremental manner. In the regression stage, the convolution, BN, ReLU, and flatten operations were conducted, as shown in Fig 5. The inputs of the regression model were the resized MBCs of the ROIs with a size of 64×64×64, which consisted of the frontal gyrus, hippocampus, amygdala, caudate nucleus, thalamus, as indicated by the orange cubes in the Fig 5. The dropout rate was set to 0.5.

Convolution with a kernel size of $3 \times 3 \times 3$ was applied, resulting in feature maps with a size of $64 \times 64 \times 64$ and 10 channels. By performing the same set of operations, feature maps with a size of $64 \times 64 \times 64 \times 20$ were obtained. Further, feature maps with a size of $64 \times 64 \times 64 \times 1$ were obtained, as indicated by the purple cube in Fig 5.

After flattening, the flattened feature map was passed through a neural network. The dimensions of the three fully connected layers were 262,144, 4096, and 64, respectively. Finally, the mapping of the fluid intelligence scores from the regressions was completed.

## 4 Experiment

In this section, we first present materials and experimental settings used in our study. We then present the quantitative evaluation metrics for segmentation results.

### A Experimental settings

In this work, the segmentation and regression components in this framework were respectively trained. The segmentation model is trained at first and after completed training the regression model is trained based on the segmented ROIs. In the first step, the improved 3D U-net is trained. We compare our proposed improved 3D U-net method with the conventional counterparts in the experiments. In the second step, the CNN is trained for regression of fluid intelligence score. We compare the CNN method with the conventional machine learning method.

Model training was carried out with 10 RTX 2080ti 11GB GPUs. The unwanted *outermost* pixels of raw data with a size of $240 \times 240 \times 240$ were removed as $224 \times 224 \times 224$ pixels, in which the outer layers of the raw data volumes were the background information. Due to the GPU memory limitation, a patch size of $112 \times 112 \times 112$ was adopted, and the batch size was 10. The patch was randomly selected from the training data, and each epoch set to 200 iterations, i.e., $200 \times 10$ patches were effectively selected in each epoch. In the segmentation stage and regression stage, we have resampled the training, test, and validation sets 5 times separately, and performed the same training and testing procedures for each resampled data.

## B. Quantitative evaluation metrics

To evaluate the performance of our segmentation approach compared to the counterparts in the experiments, we implemented the following evaluation metrics [31]. We use the Dice coefficient (DC) as the first evaluation metric. The dice coefficient is defined as the region-based similarity between the segmentation result B and the ground truth A, which can be written as

$$DC = \frac{2|A \cap B|}{|A| + |B|}, \tag{4}$$

where $|A \cap B|$ denotes the overlapped region between A and B, and $|A| + |B|$ denotes the union region.

Meanwhile, the average surface distance (ASD) also is used to measure the performance of different segmentation algorithms, which can be written as

$$ASD(A, B) = \frac{\sum_{a \in A} min_{b \in B} d(a, b)}{|A|}, \tag{5}$$

Where $d(a, b)$ is the Euclidean distance between the points of a and b, $a$ and b are the numbers of vertices in the surface A and B, respectively.

We use the mean square error (MSE) as the our CNN method and machine learning methods evaluation metric. In statistics, the MSE is defined as average of the square of the difference between true and predicted value, which can be written as

$$MSE = \frac{1}{N} \sum_{i=1}^{N} (y - y*)^2, \tag{6}$$

where, N is the total number of subjects, $y$ is the true intelligence score, $y^*$ is the predicted score from the prediction model.

## 5 Results

The proposed approach was compared with four recently proposed methods about using brain MRIs to predict fluid intelligence scores, as listed in Table 2. In comparison with the other methods, the proposed method achieved good performance with MSE = 60.29 on the training set, MSE = 51.72 on the validation set, and MSE = 82.56 on the test set.

Regarding the method proposed by Neil P. Oxtoby et al. [32], the structure and function of some brain regions were relatively tightly coupled. The structural covariance network (SCN) of the cerebral cortex was extracted, and the nodes were used as the input of the support vector regression (SVR) model to predict the intelligence score. Yeeleng S. Vang et al. [33] trained a CNN to compress 3D MRI data to a feature map size of $123 \times 1 \times 1 \times 1$, and used the 123

**Table 2. The comparison of the MSE values of the proposed method and current state-of-the-art methods.**

| Method | Train: MSE | Val: MSE | Test: MSE | References |
|---|---|---|---|---|
| SVR | 85.82 | 71.19 | 93.83 | [32] |
| CNN + GBM | 18.44 | 68.79 | 96.18 | [33] |
| 3D ConvNets | 79.28 | 70.58 | 92.74 | [13] |
| 3D U-net | - | 71.57 | 102.25 | [34] |
| **Our method** | 60.29 (53.12, 67.46) | 51.72 (45.95, 57.49) | 82.56 (75.75, 89.37) | - |

Note:—denotes the result is not reported; ( , ) denotes the upper bound of the confidence interval lower bound of confidence interval.

extracted features to train a gradient boosting machine (GBM) that predicts the intelligence score of the subject. In the experiment conducted in this study, the model performed well on the training set, but it performed poorly on the test set. To a certain extent, the model was in a state of overfitting, and was found to lack generalization ability. In the method proposed by Yukai Zou et al. [13], multiple brain regions were selected to predict the residualized fluid intelligence scores using a 3D CNN, but the median predicted score was used as the final prediction result, which caused a substantial amount of information loss in the pre-processing stage. Moreover, the regression operation of the fluid intelligence scores was performed for only one ROI at a time, which ignored the interaction between other brain areas. Lihao Liu et al. [34] used a basic 3D U-Net to enhance the segmentation performance. In this experiment, the weights of the encoder component were fixed, and the regression component was updated using the brain volume and the provided intelligence score.

From the preceding discussion, it is evident that the current models are not further optimized in terms of their algorithms and mechanisms; instead, only the original model is used. This is also the main reason for the poor performance of the prediction results. The results of the experiments and the comparisons with the existing methods demonstrate the advantages of the proposed method, which are mainly reflected in the following aspects. The current pipeline of the regression of brain MRI images and fluid intelligence scores is based on machine learning, 3D ConvNets, and 3D U-Net. However, 3D ConvNets is characterized by the following weaknesses. First, in the pre-processing stage, the high-layer and low-layer features of the 3D ConvNets framework are not fused. As a result, a large amount of spatial information in the image is lost in the lower layer. Second, in the post-processing stage, only uses feature maps with machine learning models predict scores as final prediction result. For the basic 3D U-Net, only the bottom features are used for regression with the intelligence scores.

Our method has achieved relatively good results in predicting fluid intelligence scores, mainly due to the following contributions: First, the 14 candidate ROIs are marked as five categories, i.e., when performing pixel classification, the ROIs of the same category are divided into the same label. In the segmentation task, the fewer the categories of segmentation targets, the higher the final segmentation accuracy. Second, the symmetric learning mechanism and MBC operation are beneficial to improve the prediction accuracy. Third, the many improvements made to the original 3D U-Net continuously strengthen the attention mechanism of the model, and contribute to better segmentation accuracy. Finally, in the second stage, the introduction of the CNN model increases the nonlinearity of the model, and is more conducive to the model fitting of fluid intelligence scores.

## 6 Discussion

In this section, we first compare our proposed method with several segmentation methods for brain ROI segmentation. Then, we study the influence of macro-symmetric ROIs given the same label on the segmentation results. Thirdly, we compare convolutional neural network for regression with Classical Machine Learning Methods. Finally, we present the limitations of this work.

### 6.1 Comparison with current deep learning methods

Examples of frontal gyrus, hippocampus, amygdala, caudate nucleus, thalamus segmentation results of FCN, U-Net, ResNet, FC DensNet and our method on the test dataset are shown in Fig 6. Table 3 shows the Dice coefficient (DC) and the average surface distance (ASD) values achieved by FCN [35], U-net [36], ResNet [27], FC Densenet [37] and our method. The

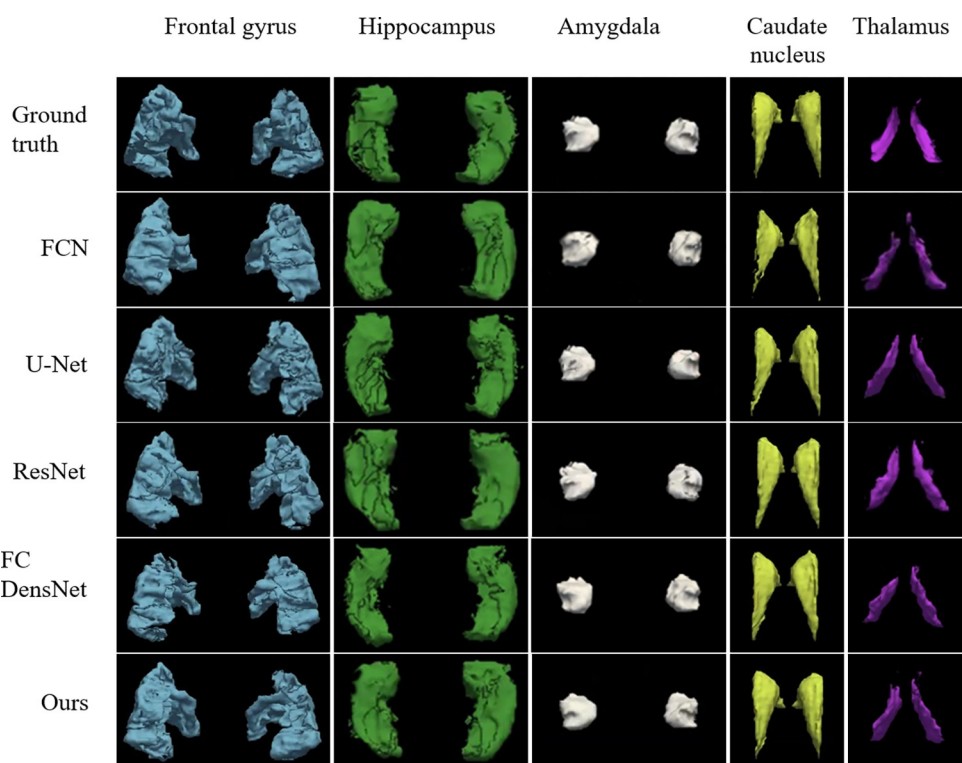

**Fig 6. Examples of frontal gyrus, hippocampus, amygdala, caudate nucleus, thalamus segmentation results of FCN, U-Net, ResNet, FC DensNet and our method on the test dataset.** The pixels in symmetrical brain regions are assigned the same label.

proposed segmentation method shows the significant improvement over the counterparts on ROIs segmentation task.

From Table 3, we can observe that the proposed method achieves the best performance for ROIs segmentation regarding the Dice coefficient metric. For example, our method achieves the highest Dice coefficient (*i.e.*, 0.8920), which is significantly better than the U-Net method (*i.e.*, 0.8394). In general, the proposed method achieves 0.0962, 0.0526, 0.0354 and 0.0298 improvement in terms of Dice coefficient over the counterparts, *i.e.*, FCN, U-Net, ResNet and FC DenseNet, respectively. Besides, the proposed method also achieves better results in terms of ASD values, compared with the counterparts. The ASD values achieved by FCN, U-Net,

**Table 3. Comparison of frontal gyrus, hippocampus, amygdala, caudate nucleus, thalamus segmentation results on the ABCD dataset.**

| Method | Dice coefficient | ASD (mm) |
|---|---|---|
| *FCN | 0.7958± 0.0257 | 0.577±0.081 |
| *U-Net | 0.8394± 0.0197 | 0.533±0.074 |
| ResNet | 0.8566±0.0256 | 0.5168± 0.059 |
| FC DenseNet | 0.8622±0.0270 | 0.496 ± 0.056 |
| Ours | 0.8920± 0.0241 | 0.390 ± 0.055 |

The terms a and b in "a ± b" denote the mean and standard deviation for different subjects, respectively. The symbol '*' indicates that the proposed method achieved significantly improvement over the other segmentation methods based on Mann Whitney U Test ($p < 0.05$) in terms of Dice coefficient.

ResNet and FC DenseNet for ROIs segmentation are 0.577, 0.533, 0.5168 and 0.496, respectively. The ASD value of our method is 0.390, which is better than other counterparts. On the other hand, compared the FCN and U-Net, the reason why U-Net performs better segmentation results is that U-Net can fuse advanced context feature information. The reason why ResNet segmentation performance is better than U-Net is that increasing the network depth can fit more complex feature inputs, and residual connection can effectively alleviate the degradation problem. FC-DenseNet has a similar structure to U-Net, adding skip connections from the encoder to the decoder and deepening the network. Therefore, FC-DenseNet has better segmentation performance than ResNet and U-Net. We can see that our method achieved the better segmentation performance than other counterparts, the possible reasons are twofold: firstly, the segmentation network is improved based on the U-Net network, which can integrate the high-level and low-level feature maps; and secondly, the concepts of ResNet and SENet are referenced to construct the network in order to suppress the degradation of the weight matrix and increase the attention mechanism, respectively [38]. We performed 5 training and testing procedures to calculate the p-value. The proposed method shows significant improvement over the FCN ($p = 0.00023$) and U-Net ($p = 0.0229$) in terms of Dice coefficient on the ABCD dataset for brain segmentation, respectively. Besides, the p-values of the proposed method over ResNet and FC DenseNet are 0.144759 and 0.202517293 in terms of Dice coefficient on the ABCD dataset for brain segmentation.

## 6.2 Influence of symmetry learning mechanism

We segmented 14 ROIs in the ABCD dataset to validate our proposed symmetry learning mechanism, where each ROI has a separate label. The segmentation results achieved by different methods are shown in Table 4. Dice coefficient and ASD are still used as the evaluation metric for our method and the counterparts.

From Table 4, we can see that the average Dice coefficient on 14 ROIs are 0.7632, 0.8084, 0.8432 and 0.8505 yielded by the counterparts, respectively, which are lower than that achieved by the proposed method (0.8686). The achieved average surface distance on 14 ROIs is 1.049 mm by our proposed method, compared with 1.225 mm, 1.214 mm, 1.119 mm, and 1.063 mm by FCN, U-Net, ResNet and FC DensNet, respectively. From Table 3, we can see that the FCN, U-Net, ResNet, FC DenseNet and the proposed method achieved 0.0326, 0.031, 0.0134, 0.0117 and 0.0234 improvements in terms of Dice coefficient over the segmentation results in Table 4, respectively. The proposed method is superior to traditional counterparts in the

**Table 4. Comparison of L inferior frontal gyrus—opercular, L inferior frontal gyrus–triangular, L inferior frontal gyrus—orbital, L hippocampus, L amygdala, L caudate nucleus, L thalamus, R inferior frontal gyrus–opercular, R inferior frontal gyrus–triangular, R inferior frontal gyrus—orbital, R hippocampus, R amygdala, R caudate nucleus, R thalamus segmentation results on the ABCD dataset.**

| Method | Dice coefficient | ASD (mm) |
|---|---|---|
| *FCN | 0.7632 ±0.0244 | 1.225 ± 0.058 |
| *U-Net | 0.8084 ± 0.0254 | 1.214 ± 0.063 |
| ResNet | 0.8432 ± 0.0310 | 1.119 ± 0.057 |
| FC DenseNet | 0.8505 ± 0.0280 | 1.063 ± 0.055 |
| Ours | 0.8686 ± 0.0236 | 1.049 ± 0.053 |

L/R indicates a location in the left/right hemisphere. The pixels in symmetrical brain regions are not assigned the same label. The terms a and b in "a ± b" denote the mean and standard deviation for different subjects, respectively. The symbol '*' indicates that the proposed method achieved significantly improvement over the other methods based on Mann Whitney U Test (p < 0.05).

separation markings in two different experiments. Also, we performed 5 training and testing procedures to calculate the p-value. The proposed method shows significant improvement over the FCN ($p$ = 2.594$e$-05) and U-Net ($p$ = 0.0196) in terms of Dice coefficient on ABCD dataset for brain segmentation.

It should be noted that the two experiments used the same experimental data. As shown in Table 3, considering that the ROI exhibits macroscopic symmetry, the pixels in the symmetrical brain region are assigned the same label during the segmentation process, and a better segmentation result is achieved. The segmentation results in Table 4 do not use a symmetric learning mechanism, one per ROI is assigned by one label.

These results demonstrate that incorporating the anatomical prior into networks could further improve the performance for brain ROI segmentation. The possible reason for the improvement is that symmetrical brain regions sharing the same label can provide more brain anatomical information for ROI segmentation, and deep neural networks can learn image features, which boost the segmentation performance around the ROI boundary.

## 6.3 Comparison with classical machine learning methods

Traditional regression methods have been widely studied in the field of predicting a continuous variable from a set of features. The support vector machine (SVM) [39], random forests (RF) [40] and gradient boosting machine (GBM) [41] are applied to predict fluid intelligence score in comparison with the proposed CNN method. In this section, we study the influence of the segmented ROIs preprocessed by three different ways on the fluid intelligence score prediction.

*1) MBCs after dimensionality reduction.* The inputs of the classical regression model were the five MBCs with a size of 64×64×64. After flattening, the dimension of flattened feature map is 262,144×5. The high dimensional data is often not useful to regression analysis [42]. Therefore, principal component analysis (PCA) is used to reduce the data dimension to 256. The input of the three machine learning models are vectors with a dimension of 256×5. We train the convolutional neural network (CNN) model using vectors with a dimension of 262,144×5.

From the Table 5, we can observe that the proposed CNN method achieves the best performance MSE = 82.56 on the test set. In comparison with the SVM, GB and RF methods, the proposed CNN method achieves 26.87, 22.91 and 21.3 improvement on the test set, respectively. The proposed CNN method shows the significant improvement over the traditional counterparts on prediction fluid intelligence score task. The possible reason for the improvement is that the CNN can adaptively learn the spatial hierarchy of low- to high-level features [42]. Besides, the proposed method shows significant improvement over the SVM

**Table 5. The comparison of the MSE values of the proposed CNN method and classical machine learning methods.**

| Method | Train: MSE | Val: MSE | Test: MSE |
|---|---|---|---|
| *SVM | 91.79 (83.06, 100.52) | 78.33 (68.69, 87.97) | 109.43 (98.8, 120.06) |
| *GB | 85.75 (77.45, 94.05) | 71.88 (64.89, 78.87) | 105.47 (96.86, 114.08) |
| *RF | 83.24 (75.30, 91.18) | 70.84 (63.94, 77.74) | 103.86 (96.37, 111.35) |
| **Our method** | 60.29 (53.12, 67.46) | 51.72 (45.95, 57.49) | 82.56 (75.75, 89.37) |

Note: ( , ) denotes the upper bound of the confidence interval lower bound of confidence interval.

The symbol '*' indicates that the proposed method achieved significantly improvement over the other methods based on Mann Whitney U Test (p < 0.05).

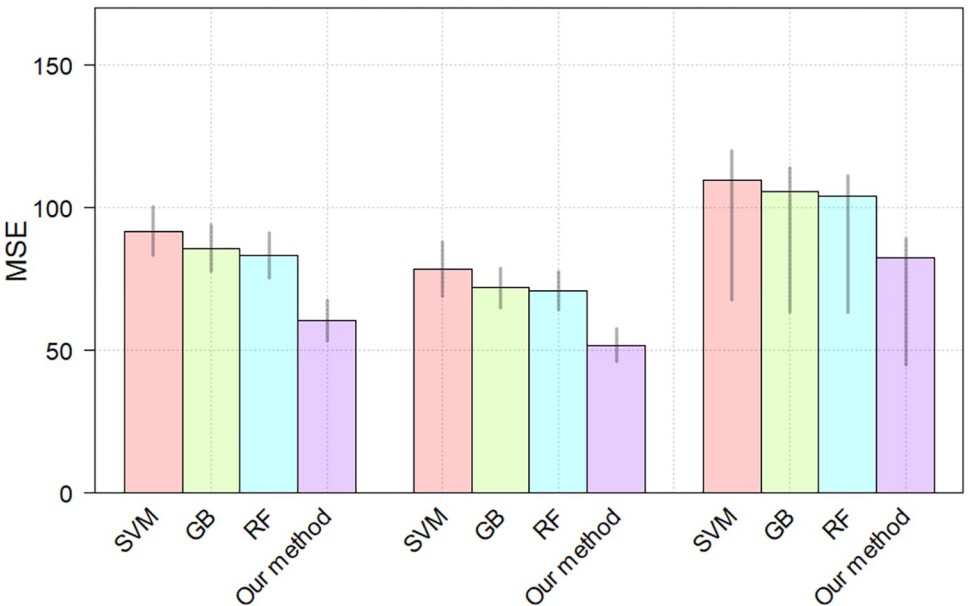

**Fig 7. The MSEs of different methods.** The heights of the bars denote the means, gray lines denote the 95% confidence intervals.

(p = 2.0205e-06), GB (p = 3.2441e-06) and RF (p = 2.694e-06) for the fluid intelligence score prediction, respectively. Fig 7 shows the MSEs of each model on the training set, validation set and test set. we performed 5 training and testing procedures to estimate the mean and 95% confidence interval of the results.

*2) ROIs after dimensionality reduction*. We now compare the influence of MBC operation and traditional resizing operation on the fluid intelligence score prediction. The segmented ROIs are resized to a size of 64x64x64 by the traditional resizing operation. The PCA is used to reduce the dimension of the segmented ROI with size of 64×64×64 to 256. Therefore, the input of the three machine learning models are vectors with a dimension of 256×5. We train the convolutional neural network (CNN) model using vectors with a dimension of 262,144×5.

From the Table 6, we can observe that the proposed CNN method achieves the best performance MSE = 86.41 on the test set. In comparison with the Table 5, the models trained with the ROI resized by the traditional method performs slightly worse. The possible reason for the slightly better performance shown in Table 5 is that MBC operation (two interpolation

**Table 6. The comparison of the MSE values of the proposed CNN method and classical machine learning methods.**

| Method | Train: MSE | Val: MSE | Test: MSE |
|---|---|---|---|
| *SVM | 93.25 (84.43, 102.07) | 78.03 (68.11, 87.95) | 110.08 (99.6, 120.56) |
| *GB | 89.46 (81.39, 97.53) | 72.75 (65.01, 80.49) | 108.36 (99.82, 116.9) |
| *RF | 82.37 (74.55, 90.19) | 74.64 (68.06, 81.22) | 105.56 (97.92, 113.2) |
| **Our method** | 62.63 (55.33, 69.93) | 54.37 (48.4, 60.34) | 86.41 (79.47, 93.35) |

Note: ( , ) denotes the upper bound of the confidence interval lower bound of confidence interval.

The symbol '*' indicates that the proposed method achieved significantly improvement over the other methods based on Mann Whitney U Test (p < 0.05).

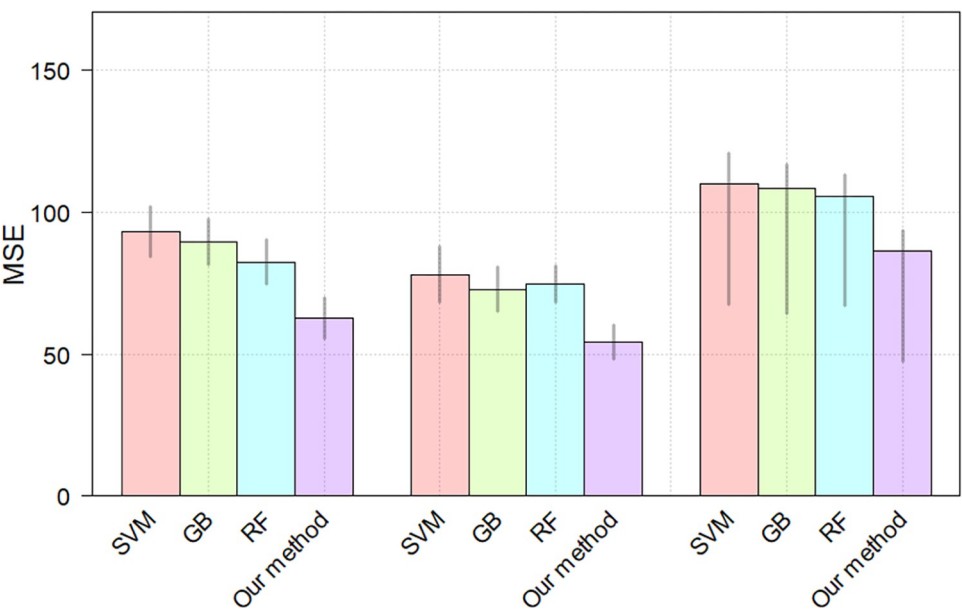

**Fig 8. The MSEs of different methods.** The heights of the bars denote the means, gray lines denote the 95% confidence intervals.

operations) produces a smoother interpolation than traditional one interpolation operation, improving the orderlity of the data, the MBC details are described in the Section 3.2—B—1). Meanwhile, the proposed method shows significant improvement over the SVM ($p = 2.8675e$-05), GB ($p = 9.0128e$-06) and RF ($p = 3.8012e$-05) for the fluid intelligence score prediction, respectively. Fig 8 shows the MSEs of each model on the training set, validation set and test set. we performed 5 training and testing procedures to estimate the mean and 95% confidence interval of the results.

*3) MBCs without dimensionality reduction.* We now study the influence of data dimensionality on model performance. The MBCs used for model training is not dimensionally reduced. The dimensions of one MBC are 262,144, details are described in the Section 3.2—B—2). We train the convolutional neural network (CNN) model and machine learning models using vectors with a dimension of 262,144×5.

From Table 7, the worse experimental results are observed compared to the Table 5. Also, the proposed method shows significant improvement over the SVM ($p = 7.1721e$-07), GB ($p = 5.4398e$-06) and RF ($p = 1.0912e$-05) for the fluid intelligence score prediction, respectively. Fig 9 shows the MSEs of each model on the training set, validation set and test set. we performed 5 training and testing procedures to estimate the mean and 95% confidence interval of the results.

**Table 7. The comparison of the MSE values of the proposed CNN method and classical machine learning methods.**

| Method | Train: MSE | Val: MSE | Test: MSE |
|---|---|---|---|
| *SVM | 101.64 (92.92, 110.36) | 97.86 (88.38, 107.34) | 127.34 (117.04, 137.64) |
| *GB | 97.86 (89.25, 106.47) | 92.53 (84.89, 100.17) | 123.55 (114.12, 132.98) |
| *RF | 92.58 (84.97, 100.19) | 90.68 (84.01, 97.35) | 122.55 (113.2, 131.9) |
| **Our method** | 74.26 (67.4, 81.12) | 67.24 (59.89, 74.59) | 95.37 (85.28, 105.46) |

Note: ( , ) denotes the upper bound of the confidence interval lower bound of confidence interval.

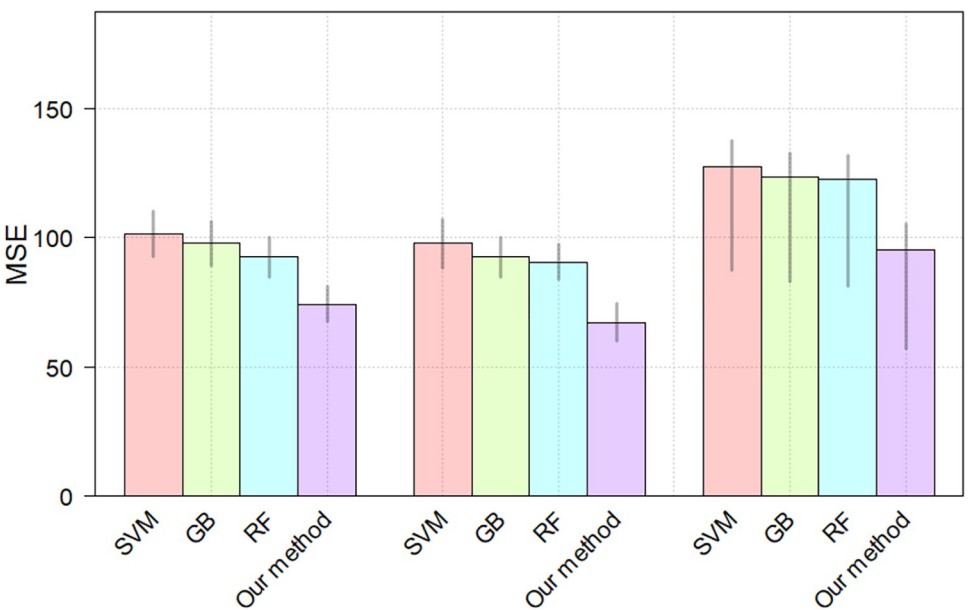

**Fig 9. The MSEs of different methods.** The heights of the bars denote the means, gray lines denote the 95% confidence intervals.

In theory, a higher number of dimensions allows more information to be stored, and there is a higher possibility of noise and data redundancy, which is not conducive to model training. As a powerful and general dimensionality reduction algorithm, PCA can unearth potential trends in our data. Therefore, using PCA for dimensionality reduction is more conducive to the convergence of the model before training the model with high-dimensional data.

### 6.4 Limitations

The proposed method was inspired by the basic 3D U-Net framework, the main concept of which is to consider the holistic perspective of intelligence predictions obtained from multiple ROIs. State-of-the-art results of multiple brain regions were achieved simultaneously. Subsequently, the fluid intelligence scores were predicted based on the fine segmentation results, which eliminated a large amount of interference and yielded more accurate and stable results. The proposed framework can be generalized to other related regression problems.

However, the SRI24 atlas is an MRI-based atlas of normal adult human brain anatomy. The participants in the ABCD project were aged between 9–10 years old, and differences in age may lead to deviations in anatomical structure matching; this is also an important factor that affected the fluid intelligence score prediction accuracy. Second, the complexity of the brain is still not fully understood, and the functional areas of the brain are quite complex. Only a few selected brain regions were used to train the model to verify the feasibility of the proposed method. Finally, the proposed method was found to achieve good results in the prediction of fluid intelligence scores. In subsequent research, the model will be further optimized while considering more brain regions to further improve the prediction accuracy.

### 7 Conclusion

In this paper, a two-step deep learning pipeline was proposed to predict fluid intelligence scores from T1-w MRI images. In the first step, an improved 3D U-Net was trained to segment the 3D MRI data to obtain the target brain areas. The proposed architecture is a combination

of ResNet, SENet, and the symmetry learning mechanism to increase the segmentation accuracy. In the second step, a CNN was trained to predict the fluid intelligence scores based on the fine segmentation results. Compared with the current state-of-the-art methods for the prediction of fluid intelligence scores from T1-weighted MRI images, the proposed method includes the addition of different modules to improve the attention mechanism of the entire model, thereby contributing to better prediction results. The proposed framework can be validated and improved in the future, and it offers a new and unique perspective for the prediction of fluid intelligence scores based on brain morphometry.

## Supporting information

**S1 File. Fluid intelligence score measurement.**
(DOCX)

## Author Contributions

**Conceptualization:** Mingliang Li.

**Data curation:** Mingliang Li, Guangming Zhang.

**Methodology:** Mingliang Li, Mingfeng Jiang.

**Software:** Mingliang Li, Mingfeng Jiang, Yujun Liu.

**Supervision:** Guangming Zhang, Xiaobo Zhou.

**Validation:** Mingliang Li.

**Writing – original draft:** Mingliang Li.

**Writing – review & editing:** Mingliang Li.

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
