## [Decision Letter · Decision Letter 0]

13 Jan 2022

PONE-D-21-37778Prediction of fluid intelligence from T1-w MRI images: A precise two-step deep learning frameworkPLOS ONE

Dear Dr. Li,

Thank you for submitting your manuscript to PLOS ONE. After careful consideration, we feel that it has merit but does not fully meet PLOS ONE’s publication criteria as it currently stands. Therefore, we invite you to submit a revised version of the manuscript that addresses the points raised during the review process.

We look forward to receiving your revised manuscript.

Kind regards,

Yiming Tang, Ph.D.

Academic Editor

PLOS ONE

Journal Requirements:

[This research was supported by Center of Excellence-International Collaboration Initiative Grant (Grant Number: 139170052), 1.3.5 project for disciplines of excellence, West China Hospital, Sichuan University (Grant Number: ZYJC18010).]

 [This research was supported by Center of Excellence-International Collaboration Initiative Grant (Grant Number: 139170052), 1.3.5 project for disciplines of excellence, West China Hospital, Sichuan University (Grant Number: ZYJC18010). ]

5. PLOS requires an ORCID iD for the corresponding author in Editorial Manager on papers submitted after December 6th, 2016. Please ensure that you have an ORCID iD and that it is validated in Editorial Manager. To do this, go to ‘Update my Information’ (in the upper left-hand corner of the main menu), and click on the Fetch/Validate link next to the ORCID field. This will take you to the ORCID site and allow you to create a new iD or authenticate a pre-existing iD in Editorial Manager. Please see the following video for instructions on linking an ORCID iD to your Editorial Manager account: https://www.youtube.com/watch?v=_xcclfuvtxQ.

Reviewers' comments:

Reviewer's Responses to Questions

**Comments to the Author**

1. Is the manuscript technically sound, and do the data support the conclusions?

Reviewer #1: Partly

Reviewer #2: Partly

2. Has the statistical analysis been performed appropriately and rigorously? 

Reviewer #1: No

Reviewer #2: No

3. Have the authors made all data underlying the findings in their manuscript fully available?

Reviewer #1: Yes

Reviewer #2: No

4. Is the manuscript presented in an intelligible fashion and written in standard English?

Reviewer #1: No

Reviewer #2: Yes

5. Review Comments to the Author

Reviewer #1: Thank you very much for the opportunity to review the manuscript entitled "Prediction of fluid intelligence from T1-w MRI images: a precise two-step deep learning framework".

Although the methodology seems fine in general, there remain open questions on what the authors did exactly. During reading I had the impression that the authors had some difficulties with the English language and structuring the text in an intelligible manner. This made it not only difficult for me to follow sometimes, but also led to wrong statements. One example for this is the last sentence in the 2nd paragraph of the Introduction: "In the most related work, MRI data were used as a machine learning method by which to predict fluid intelligence." This statement implies that MRI data is a machine learning method, however, MRI data itself is NOT a machine learning method. Machine learning methods are APPLIED TO MRI data. Sentences like the one mentioned unfortunately worsen the overall impression of the manuscript.

In the following I will list some points, which would in my opinion improve the manuscript:

- Introduction 5th paragraph: The drawbacks that are listed are not drawbacks of the MRI data itself but of the methodology applied to it, so the first sentence should be reformulated. It should also be pointed out in what terms the segmentation model is not well optimized in previous work.

- Section 2.1: The structure of the section is a bit strange, for me it seems as if paragraphs had been randomly shuffled. For me it would make more sense to start the section with the first sentence of the 3rd paragraph. Then it should be described what processing has been performed by the challenge organizers and after that the preprocessing steps done by the authors. Furthermore, the last sentence of the section is the same as the one forming the second paragraph.

- Section 2.2 Please also mention somewhere in the text explicitly that the ROIs are assigned to five categories. Please also describe how the ground truth for the parcellation was created. Was the parcellation provided through the challenge or a freely available parcellation used, etc.?

- Section 3.2 A, paragraph 4: The description of the first layer does not seem to fit to Fig. 2. In figure the dimensions change 112*112*112*112*1 112*112*112*16  112*112*112*32, but in the text it is unclear if it is 112*112*112*112*1 112*112*112*32  112*112*112*12, or 112*112*112*112*1 112*112*112*12  112*112*112*32. Also note that in the image there are 16 channels vs. 12 channels mentioned in the text.

- Section 3.2. A1 Recombination Block: In the text it is said that the operation from B2 to B3 is a 1x1x1 conv, however in the image it is a 3x3x3 conv.

-Section 3.2. A2 SegS-E Block: In the 3rd line below Fig 4, the output is denoted as Y but later the output is denoted as X tilde

- Section 3.2 B1 Minimum bounding cube: This section is very confusing. It seems that there is only 1 sentence on what the minimum bounding cube is and the rest of the 2 paragraphs is about model training. I would suggest to add a separate section on model training after the description of the whole framework and focus in this section only on the MBC. Actually, I do not really understand what the MBC exactly is and what exactly the input to the neural network for regression is.

Concerning model training: It is not totally clear for me if the segmentation and regression component are trained together or if the segmentation component is trained at first and after completed training the regression component is trained based on the results of the segmentation component.

- Section 3.2 B2 Neural network construction: In the last sentence "onto the ROIs" should be deleted.

Results and Discussion in general: I would suggest to put all experimental results (especially the tables) into the results section and every interpretation of results (why one method might be better than another, etc) into the discussion section.

Table 2: If possible, it would be nice to have mean and standard deviation over several training runs/ seeds reported, instead of just a value for training each method only once.

Section 5.1: Please check the grammar of the first sentence.

Table 3 and 4: Pleas provide mean and standard deviation over several training runs instead just 1.

Section 5.3: I would like to see also the results of SVM, RF and GB based on the higher dimensional data. Since the CNN has more information available than the other approaches, this seems to be a bit of an unfair comparison. By reducing the dimension with PCA you might discard information that is useful for prediction, since the components corresponding to highest variance, do not necessary have to reflect also the most useful information. How much variance the kept components explain would also be interesting to know.

The authors also claim that the CNN shows SIGNIFICANT improvement over the other methods. However, Table 5 includes only MSE-values for training the CNN and other methods only once. p-values or confidence intervals have to be added to provide evidence for significant improvement.

Reviewer #2: Major remark :

1) We know that the quality of measurement of intelligence is linked to validity and can

interfere on results of each study (see for example, Gignac et al. (2017, https://doi.org/10.1016/j.intell.2017.06.004). The authors do not give any indication of the measurement of fluid intelligence. This information, although necessary, is difficult to obtain from the link https://nda.nih.gov/abcd/about. and requires an in-depth analysis of the database from which the data were extracted (NIMH Data Archive (NDA) database, ABCD study). This is not within the reach of all readers. A summary table of the measurements and tests used will certainly be useful.

Minor remarks :

2) The authors used the MSE value to compare the proposed CNN method to the classical machine learning methods. Choosing MSE is a good choice since it is a well-behaved metric, but correlation criteria should also be provided because the correlation score is used in many other studies on fluid intelligence prediction.

3) Results need to be supported by statistical analysis showing at what extent the differences between the proposed method and the other methods are significant.

4) In p. 3 : Fluid intelligence scores were decidualized ( ??). is « decidualized » the right word ?.

5) In p. 13: « The ROIs of frontal gyrus, hippocampus, amygdala, caudate nucleus and

thalamus are performed segmentation ». Should be rephrased.

6) In Fig.6, it is not clear that the proposed method shows significant improvement

over the counterparts on ROIs segmentation task

7) p.17, « adaptively learn » is repeated twice. Please correct.

8) Caption table 4 : the word « sementation » is repeated twice. Please correct.

6. PLOS authors have the option to publish the peer review history of their article (what does this mean?). If published, this will include your full peer review and any attached files.

Reviewer #1: No

Reviewer #2: **Yes: **Abdel-Kader Boulanouar

---

## [Author Response · Author response to Decision Letter 0]

29 Mar 2022

Reviewer #1: Thank you very much for the opportunity to review the manuscript entitled "Prediction of fluid intelligence from T1-w MRI images: a precise two-step deep learning framework".

Although the methodology seems fine in general, there remain open questions on what the authors did exactly. During reading I had the impression that the authors had some difficulties with the English language and structuring the text in an intelligible manner. This made it not only difficult for me to follow sometimes, but also led to wrong statements. One example for this is the last sentence in the 2nd paragraph of the Introduction: "In the most related work, MRI data were used as a machine learning method by which to predict fluid intelligence." This statement implies that MRI data is a machine learning method, however, MRI data itself is NOT a machine learning method. Machine learning methods are APPLIED TO MRI data. Sentences like the one mentioned unfortunately worsen the overall impression of the manuscript.

In the following I will list some points, which would in my opinion improve the manuscript:

Q1- One example for this is the last sentence in the 2nd paragraph of the Introduction: "In the most related work, MRI data were used as a machine learning method by which to predict fluid intelligence." This statement implies that MRI data is a machine learning method, however, MRI data itself is NOT a machine learning method. Machine learning methods are APPLIED TO MRI data. Sentences like the one mentioned unfortunately worsen the overall impression of the manuscript.

Reply: It is really a big mistake to the whole quality of our article. We feel sorry for our carelessness. We have corrected it and we also feel great thanks for your point out. The corrected statemen “In the most related work, the reference [4] has outlined machine learning approaches employed to predict fluid intelligence from brain MRI data.” has been placed in the last sentence of 2nd paragraph of the Introduction.

Q2 - Introduction 5th paragraph: The drawbacks that are listed are not drawbacks of the MRI data itself but of the methodology applied to it, so the first sentence should be reformulated. It should also be pointed out in what terms the segmentation model is not well optimized in previous work.

Reply: Thanks for your suggestions. We have revised the first sentence of the fifth paragraph, and also given what the segmentation model needs to be optimized.

Q3 - Section 2.1: The structure of the section is a bit strange, for me it seems as if paragraphs had been randomly shuffled. For me it would make more sense to start the section with the first sentence of the 3rd paragraph. Then it should be described what processing has been performed by the challenge organizers and after that the preprocessing steps done by the authors. Furthermore, the last sentence of the section is the same as the one forming the second paragraph.

Reply: The content of section 2.1 in the original manuscript appears somewhat illogical. According to your comments, we have made a reformulated revision and placed it in the 1st paragraph of section 2.1.

In the 2nd paragraph of section 2.1, it is described what preprocessing has been performed by the challenge organizers. Preprocessing includes format conversion, selection of the brain mask, and creation of the parcellation. The challenge organizers performed brain parcellation using neuroimaging software. Finally, 122 brain regions of interest (ROIs) extracted by the challenge organizers based on the SRI24 atlas.

The 3rd paragraph of section 2.1 describes the preprocessing work performed by the authors. 14 brain ROIs with unique anatomical characteristics and roles in cognitive functions were selected to predict fluid intelligence score.

Q4- Section 2.2 Please also mention somewhere in the text explicitly that the ROIs are assigned to five categories. Please also describe how the ground truth for the parcellation was created. Was the parcellation provided through the challenge or a freely available parcellation used, etc.?

Reply: In the 2nd paragraph of Section 2.2, we have given the categories to which ROIs are assigned. We put the description of the ground truth for the parcellation in the 2nd paragraph of Section 2.1 because creating parcellation is the focus of preprocessing.

 The challenge organizers created brain parcellation using neuroimaging software, the details are put in the 2nd paragraph of section 2.1.Finally, 122 brain regions of interest (ROIs) extracted by the challenge organizers based on the SRI24 atlas.

Q5- Section 3.2 A, paragraph 4: The description of the first layer does not seem to fit to Fig. 2. In figure the dimensions change 112*112*112*112*1 112*112*112*16  112*112*112*32, but in the text it is unclear if it is 112*112*112*112*1 112*112*112*32  112*112*112*12, or 112*112*112*112*1 112*112*112*12  112*112*112*32. Also note that in the image there are 16 channels vs. 12 channels mentioned in the text.

Reply: We have re-drawn the Fig 2. We have revised the description of the first layer in the first two sentences of paragraph 4, section 3.2 A. Thank you for your detailed review! Yes, in the image there are 16 channels, no 12 channels.

Q6- Section 3.2. A1 Recombination Block: In the text it is said that the operation from B2 to B3 is a 1x1x1 conv, however in the image it is a 3x3x3 conv.

Reply: we have given the revised it, and placed it in the 3rd sentence of last paragraph in Section 3.2 -A - 1).

Q7-Section 3.2. A2 SegS-E Block: In the 3rd line below Fig 4, the output is denoted as Y but later the output is denoted as X tilde

Reply: I am sorry, ‘Y’ is a typo. We have put the X tilde in the 3rd line below Fig 4.

Q8- Section 3.2 B1 Minimum bounding cube: This section is very confusing. It seems that there is only 1 sentence on what the minimum bounding cube is and the rest of the 2 paragraphs is about model training. I would suggest to add a separate section on model training after the description of the whole framework and focus in this section only on the MBC. Actually, I do not really understand what the MBC exactly is and what exactly the input to the neural network for regression is.

Concerning model training: It is not totally clear for me if the segmentation and regression component are trained together or if the segmentation component is trained at first and after completed training the regression component is trained based on the results of the segmentation component.

Reply: we have focused only on the MBC, and we have re-described what MBC is in Section 3.2 B-1). In this work, the prediction of fluid intelligence scores in two stages, the segmentation model is trained at first. Then, the regression model is trained using the segmented ROIs for fluid intelligence score prediction.

We have added a new Section 4 (Experiment). The Experimental Settings are described in the new Section 4 – A and the Quantitative Evaluation Metrics in the new Section 4– B.

In addition, we have compared the influence of MBC operation and traditional resizing operation on the fluid intelligence score prediction, details are shown in the new Section 6.3-2). And we have made the statistical analysis for the results.

Q9- Section 3.2 B2 Neural network construction: In the last sentence "onto the ROIs" should be deleted.

Reply: we have deleted the "onto the ROIs" In the last sentence in Section 3.2 B2 Neural network construction.

Q10-Results and Discussion in general: I would suggest to put all experimental results (especially the tables) into the results section and every interpretation of results (why one method might be better than another, etc) into the discussion section.

Reply: Thank you your suggestion. The predicted results of the fluid intelligence scores are presented in the new Section 5 (Results) as the most important results. 

 In the new Section 6 (Discussion), we have added additional experimental results, including tables and figures. After our discussion, we think that putting other results in the new Section 6 (Discussion) will be more convenient for readers to read and think.

Q11-Table 2: If possible, it would be nice to have mean and standard deviation over several training runs/ seeds reported, instead of just a value for training each method only once.

Reply: Unfortunately, the parameter settings details of the models in the references are not given, and it is difficult for us to reproduce their methods.

 In Table 2, we have given the mean and the confidence interval of our method. In addition, all the results in this paper have been supported by statistical analysis. 

In our work, in the segmentation stage and regression stage, we have resampled the training, test, and validation sets 5 times separately, and performed the same training and testing procedures for each resampled data

Q12-Section 5.1: Please check the grammar of the first sentence.

Reply: We have reformulated the first sentence in the Section 6.1 in the revised manuscript.

Q13-Table 3 and 4: Pleas provide mean and standard deviation over several training runs instead just 1. 

Reply: We have resampled the training, test, and validation sets 5 times separately, and performed the same training and testing procedures for each resampled data. We have given the new mean and standard deviation in Table 3 and 4. 

Besides, we have made the statistical analysis for the Table 3 and 4. The details are presented in the last two sentence of the last paragraph of new Section 6.1, and in the last two sentence of the 2nd paragraph of the new Section 6.2.

Q14-Section 5.3: I would like to see also the results of SVM, RF and GB based on the higher dimensional data. Since the CNN has more information available than the other approaches, this seems to be a bit of an unfair comparison. By reducing the dimension with PCA you might discard information that is useful for prediction, since the components corresponding to highest variance, do not necessary have to reflect also the most useful information. How much variance the kept components explain would also be interesting to know.

reply: We have given the result of models training using higher dimensional data as input for, details are described in the new Section 6.3- 3).

In theory, a higher number of dimensions allows more information to be stored, and there is a higher possibility of noise and data redundancy, which is not conducive to model training. The dimensionality reduction can unearth potential trends in data. Our experimental results confirm that dimensionality reduction operations are still necessary for high-latitude data before model training.

Q15-The authors also claim that the CNN shows SIGNIFICANT improvement over the other methods. However, Table 5 includes only MSE-values for training the CNN and other methods only once. p-values or confidence intervals have to be added to provide evidence for significant improvement.

Reply: We have resampled the training, test, and validation sets 5 times separately, and performed the same training and testing procedures for each resampled data. All the results have been supported by statistical analysis. P-values or confidence intervals have been provided for significant improvement.

Reviewer #2: Major remark :

1) We know that the quality of measurement of intelligence is linked to validity and can

interfere on results of each study (see for example, Gignac et al. (2017, https://doi.org/10.1016/j.intell.2017.06.004). The authors do not give any indication of the measurement of fluid intelligence. This information, although necessary, is difficult to obtain from the link https://nda.nih.gov/abcd/about. and requires an in-depth analysis of the database from which the data were extracted (NIMH Data Archive (NDA) database, ABCD study). This is not within the reach of all readers. A summary table of the measurements and tests used will certainly be useful.

Reply: Thanks for your suggestion. We have cited the reference (2017, https://doi.org/10.1016/j.intell.2017.06.004) in the 2nd sentence of the 2nd paragraph of Section 1 Introduction. We have provided a summary table of the measurements and tests Table details in supplement.

Minor remarks :

2) The authors used the MSE value to compare the proposed CNN method to the classical machine learning methods. Choosing MSE is a good choice since it is a well-behaved metric, but correlation criteria should also be provided because the correlation score is used in many other studies on fluid intelligence prediction.

Reply: We have provided the correlation criteria of MSE as as our CNN method and machine learning methods evaluation metric in the new Section 4 (Experiment) – B (Quantitative Evaluation Metrics).

3) Results need to be supported by statistical analysis showing at what extent the differences between the proposed method and the other methods are significant.

Reply: All the results have been supported by statistical analysis.

4) In p. 3 : Fluid intelligence scores were decidualized ( ??). is « decidualized » the right word ?.

Reply: Thanks for your suggestion. The “decidualized” is a wrong word. we have revised it.

5) In p. 13: « The ROIs of frontal gyrus, hippocampus, amygdala, caudate nucleus and

thalamus are performed segmentation ». Should be rephrased.

Reply: we have reformulated the first sentence of the new Section 6.1. 

6) In Fig.6, it is not clear that the proposed method shows significant improvement

over the counterparts on ROIs segmentation task

Reply: We agree with your comment. To evaluate accurately the performance of our segmentation approach compared to the counterparts in the experiments, we implemented the following evaluation metrics: Dice coefficient (DC) and the average surface distance (ASD), details are shown in the new Section 4 (Experiment) – B.

7) p.17, « adaptively learn » is repeated twice. Please correct.

Reply: We have deleted extra “adaptively learn”, as shown in the 4th sentence of the 2nd paragraph of the new Section 6.3 – 1).

8) Caption table 4 : the word « sementation » is repeated twice. Please correct.

Reply: we have deleted the extra “segmentation” in Caption table 4.

---

## [Decision Letter · Decision Letter 1]

6 May 2022

Prediction of fluid intelligence from T1-w MRI images: A precise two-step deep learning framework

PONE-D-21-37778R1

Dear Dr. Li,

We’re pleased to inform you that your manuscript has been judged scientifically suitable for publication and will be formally accepted for publication once it meets all outstanding technical requirements.

Kind regards,

Yiming Tang, Ph.D.

Academic Editor

PLOS ONE

Additional Editor Comments (optional):

Reviewers' comments:

Reviewer's Responses to Questions

**Comments to the Author**

1. If the authors have adequately addressed your comments raised in a previous round of review and you feel that this manuscript is now acceptable for publication, you may indicate that here to bypass the “Comments to the Author” section, enter your conflict of interest statement in the “Confidential to Editor” section, and submit your "Accept" recommendation.

Reviewer #1: All comments have been addressed

Reviewer #2: All comments have been addressed

2. Is the manuscript technically sound, and do the data support the conclusions?

Reviewer #1: Yes

Reviewer #2: Yes

3. Has the statistical analysis been performed appropriately and rigorously? 

Reviewer #1: Yes

Reviewer #2: Yes

4. Have the authors made all data underlying the findings in their manuscript fully available?

Reviewer #1: Yes

Reviewer #2: Yes

5. Is the manuscript presented in an intelligible fashion and written in standard English?

Reviewer #1: Yes

Reviewer #2: Yes

6. Review Comments to the Author

Reviewer #1: (No Response)

Reviewer #2: the authors have extensively modified the manuscript to correctly answer the remarks and questions of the reviewers.

7. PLOS authors have the option to publish the peer review history of their article (what does this mean?). If published, this will include your full peer review and any attached files.

Reviewer #1: No

Reviewer #2: No

---

## [Editor Report · Acceptance letter]

22 Jul 2022

PONE-D-21-37778R1 

Prediction of fluid intelligence from T1-w MRI images: A precise two-step deep learning framework 

Dear Dr. Li:

I'm pleased to inform you that your manuscript has been deemed suitable for publication in PLOS ONE. Congratulations! Your manuscript is now with our production department. 

Kind regards, 

on behalf of

Professor Yiming Tang 

Academic Editor

PLOS ONE